# Systemic lupus erythematosus favors the generation of IL-17 producing double negative T cells

Hao Li[1], Iannis E. Adamopoulos[2], Vaishali R. Moulton[1], Isaac E. Stillman[3], Zach Herbert[4], James J. Moon[5], Amir Sharabi[1], Suzanne Krishfield[1], Maria G. Tsokos[1] & George C. Tsokos[1✉]

Mature double negative (DN) T cells are a population of αβ T cells that lack CD4 and CD8 coreceptors and contribute to systemic lupus erythematosus (SLE). The splenic marginal zone macrophages (MZMs) are important for establishing immune tolerance, and loss of their number or function contributes to the progression of SLE. Here we show that loss of MZMs impairs the tolerogenic clearance of apoptotic cells and alters the serum cytokine profile, which in turn provokes the generation of DN T cells from self-reactive CD8$^+$ T cells. Increased Ki67 expression, narrowed TCR V-beta repertoire usage and diluted T-cell receptor excision circles confirm that DN T cells from lupus-prone mice and patients with SLE undergo clonal proliferation and expansion in a self-antigen dependent manner, which supports the shared mechanisms for their generation. Collectively, our results provide a link between the loss of MZMs and the expansion of DN T cells, and indicate possible strategies to prevent the development of SLE.

[1] Department of Medicine, Beth Israel Deaconess Medical Center, Harvard Medical School, Boston, MA, USA. [2] Division of Rheumatology, Allergy and Clinical Immunology, University of California, Davis, CA 95817, USA. [3] Department of Pathology, Beth Israel Deaconess Medical Center, Harvard Medical School, Boston, MA 02115, USA. [4] Molecular Biology Core Facilities, Dana-Farber Cancer Institute, 21-27 Burlington Ave, Boston, MA 02215, USA. [5] Center for Immunology and inflammatory Diseases, Massachusetts General Hospital, Harvard Medical School, Charlestown, MA 02129, USA. ✉email: gtsokos@bidmc.harvard.edu

Systemic lupus erythematosus (SLE) is a complex autoimmune disease driven, in part, by immune activation against autoantigens present on blebs of apoptotic or necrotic cells[1–3]. Considerable evidence supports the notion that systemic autoimmunity can be initiated by impaired or delayed clearance of dead cells, which leads to a global loss of self-tolerance through the activation of autoreactive T and B cells[4,5].

Among the prominent T-cell abnormalities which have been reported in patients with SLE are the expansion of pathogenic Th17[6] and TCR-αβ+CD4−CD8− double-negative (DN) T cells[7–9]. Several lines of evidence suggest that DN T cells may arise from activated CD8 T cells[10–14], but the specific factors which promote CD8 loss in patients and mice with SLE is not understood.

Reduction in the number or function of MZMs has been shown not only in several lupus-prone mice but also in SLE patients[15–19]. MZMs, expressing an array of scavenger receptors including macrophage receptor with collagenous structure (MARCO), scavenger receptor A (SR-A), and SIGN-R1[19–21], surround the splenic follicles and have been reported capable of both efficiently clearing apoptotic or necrotic cells and inducing immune tolerance[15,17,19,22]. Absence of MZMs results in retention of cellular debris derived from dead cells and provokes robust T-cell responses against self-antigens[16–18]. Here, we provide the missing link between MZM defects and DN T-cell expansion by demonstrating that self-antigens derived from uncleared cellular debris favor the conversion of self-reactive CD8 T cells into DN T cells. We further prove that MZM defects generate an inflammatory milieu, in which elevated IL-23 along with reduced TGFβ facilitate self-reactive DN T-cell activation, expansion, and survival. Because mouse and human DN T cells express Ki67, narrowed TCR V beta-repertoire usage and diluted T-cell receptor excision circles (TREC), we postulate that shared mechanisms are involved in their generation and suggest that blockade of IL-23 and/or the empowerment of TGFβ signaling should have therapeutic value.

## Results

**MZM depletion promotes DN T cells in lupus-prone mice.** Absence of MZMs results in retention of dead cell debris which drives the activation of autoreactive T and B cells in lupus-prone mice and patients with SLE[15–18]. Previous reports have shown that long-term removal of MZMs accelerates autoimmunity in lupus-prone mice[15,17]. To examine the effect of altered immune responsiveness to ACs on the development of DN T cells in the B6.lpr lupus-prone mouse, we depleted selectively MZMs in female mice in vivo by weekly injection of appropriate dose of clodronate liposomes (Clodrosome, 100 μg/mouse)[15,17,23] (Supplementary Figs. 1 and 2) beginning at 10 weeks of age. PBS-loaded control liposomes (PBSLs) were used as control. As expected, Clodrosome-mediated MZM depletion accelerated the onset of autoimmunity. When MZMs were depleted, the mice showed enhanced autoimmunity, including elevated serum anti-dsDNA titers (Fig. 1a), increased formation of spontaneous germinal centers (Fig. 1b, upper panel), and expanded IL-17-producing DN T cells in the spleens (Fig. 1b, low panel; Fig. 1c, d). Confocal microscopic images further confirmed the expansion of DN T cells in the absence of MZMs (Fig. 1e). In addition, we noted a significant increase in the numbers of DN T cells infiltrating the kidneys (Fig. 1f, g). Taken together, our results suggest that the lack of MZMs leads to the expansion of DN T cells in this lupus-prone mouse strain both in the spleen and the kidney and promotes disease development.

**Exposure to apoptotic cell debris induces CD8 loss.** In lpr mice, defects in apoptosis lead to increased rates of other forms of

programmed cell death[24,25]. To investigate whether the accumulation of uncleared dead cell debris and the presence of associated antigens are mechanistically required for the generation of DN T cell from self-reactive CD8 T cells and whether the lupus milieu marked by MZM deficiency could facilitate this conversion, we co-transferred CFSE-labeled CD8+ OT-I TCR transgenic (Tg) T cells with apoptotic thymocytes derived from m-OVA Tg mice into B6 mice treated with or without Clodrosome. In the absence of MZMs, a larger fraction of the transferred CFSE-labeled OT-I T cells lost CD8 expression and the percentage of CD8− T cells that entered the cell cycle was higher compared with those in which MZMs were present (Fig. 2a; Supplementary Fig. 3a). In accordance with our previous reports that DN T cells found in SLE patients and lupus-prone mice produced high levels of IL-17[8], newly formed DN T cells gained the ability to produce IL-17 but failed to produce IFNγ (Fig. 2a). To further confirm our findings that DN T cells originate from CD8 T cells, we isolated OT-II CD4 T cells and transferred them into B6 mice treated with or without Clodrosome. We did not observe downregulation of CD4 expression on the vigorously proliferating OT-II T cells after the transfer of apoptotic m-OVA+ thymocytes (Fig. 2b; Supplementary Fig. 4).

To investigate whether endogenous dead cell debris can efficiently promote DN T-cell generation, we transferred either OT-I or OT-II T cells into m-OVA Tg mice expressing OVA ubiquitously. Cell proliferation of OT-I (Fig. 2c; Supplementary Fig. 3b) or OT-II (Fig. 2d) T cells was noted 3 days after transfer, and a more robust proliferation was observed when the recipient mice had received Clodrosome (Fig. 2c, d). As expected, a significant higher percentage of CFSE-labeled OT-I T cells displayed a phenotype characterized by loss of CD8 expression (Fig. 2c). Accordingly, downregulation of CD4 expression was not observed on transferred OT-II cells in m-OVA Tg recipients regardless of the presence of MZMs (Fig. 2d; Supplementary Fig. 5). Loss of CD8 but not CD4 expression on transferred OT-I or OT-II T cells was further confirmed by confocal image analysis (Fig. 2e, f). Of interest, in m-OVA Tg recipients of OT-I T cells, T lymphocyte infiltration could be tracked inside the kidneys, originating from both donors and recipients (Fig. 2g). As noted, intrarenal DN T cells originated exclusively from the donors but not from the recipients (Fig. 2g), suggesting that presence of self-antigens is the requisite for DN T-cell formation and migration.

**Autoreactive polyclonal T cells preferentially lose CD8.** The limitation of monoclonal T-cell population in OT-I or OT-II TCR Tg mice and the broad T-cell repertoire diversity in different individuals brought forward the question whether the conversion from CD8 to DN T cells could really happen in SLE. To examine the effects of altered immune responsiveness to ACs on the development of DN T cells in murine models in which self-reactive CD8 T cells were enriched in either central or peripheral lymphoid organs, a mixed cell transfer experiment was carried out in which flow cytometry sorted CD8 T cells (TCRβ+NK1.1−CD8+) from Cd45.1 B6 mice were transferred along with CD8 T cells from Cd45.2 B6.lpr.DsRed (Fig. 3a) into Cd45.2 B6 recipients. The response of transferred T cells to co-transferred apoptotic thymocytes and the possible downregulation of CD8 were assessed by flow cytometry. Significantly higher numbers of transferred T cells from Cd45.2 B6.lpr.DsRed were observed in the spleens of the recipients receiving Clodrosome plus apoptotic B6 thymocytes than the spleens of matched controls or those receiving only apoptotic B6 thymocytes (Fig. 3a, left panel; Supplementary Fig. 6a). Accordingly, the more the cells proliferate the higher the CD8 loss occurs (Fig. 3a, right panel) which is consistent with the downregulation of CD8 expression on OT-I T cells upon antigen stimulation described

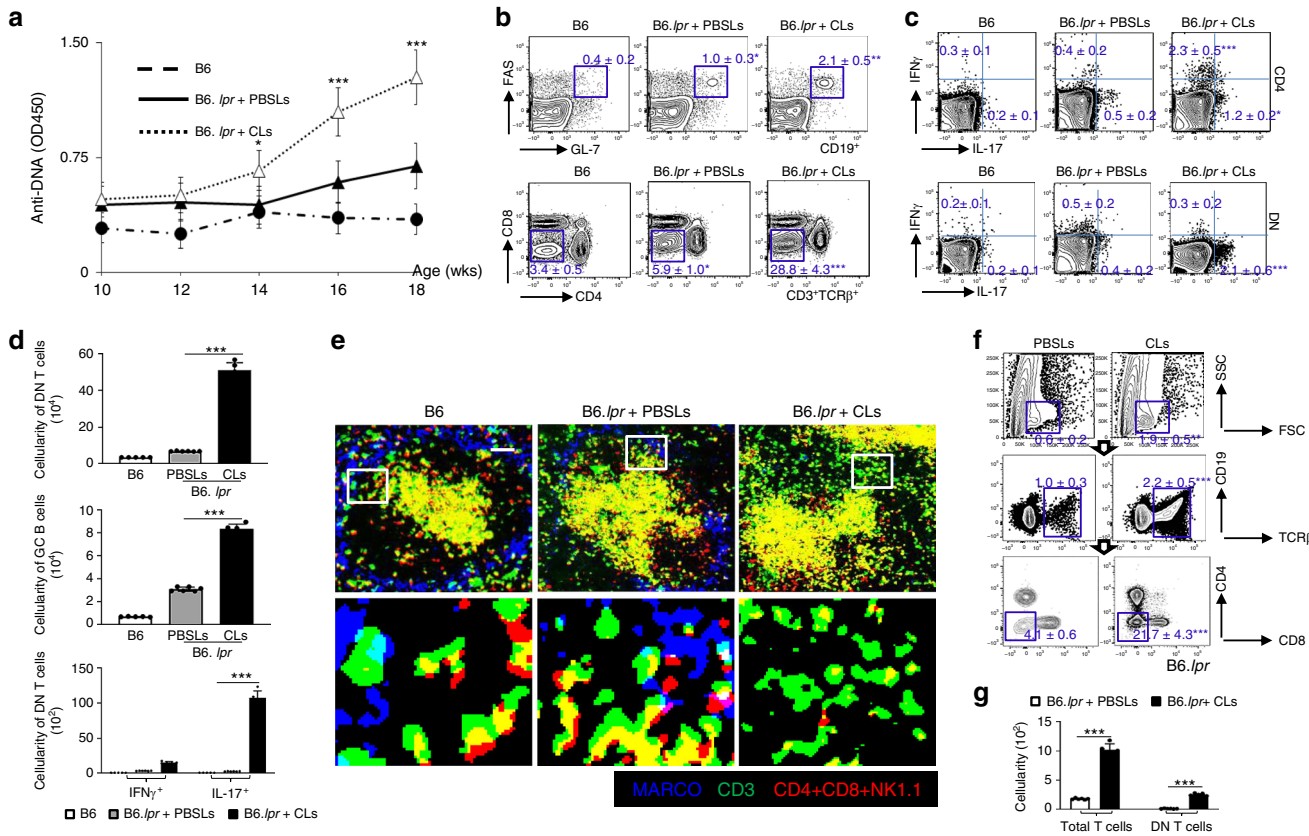

**Fig. 1 Marginal macrophage depletion promotes the expansion of DN T cells. a–g** Age-matched female B6.*lpr* mice were treated with either PBS-loaded control liposomes (PBSLs) or Clodronate liposomes (CLs, 100 μg/mouse) every other week for 2 months starting at 10 weeks of age. Naive B6 mice were used as controls. $n = 5$–6 mice per group for two independent experiments. **a** ELISA analysis of anti-dsDNA IgG in sera from indicated mice. **b** Flow cytometry quantization of the percentage of germinal center (GC) B cells and double-negative (DN) T cells in spleens from indicated mice. Upper: PNA+FAS+ GC B cells (gated in CD19+); lower: CD4−CD8−NK1.1− DN T cells (gated in CD3+TCRβ+). **c** Flow cytometry quantitation of the percentage of IL-17+ and IFN-γ+ CD4 T cells (upper) and DN T cells (lower) in spleens of indicated mice. **d** Bar graphs show absolute cell numbers of indicated cell population of spleens from mice with indicated treatment. **e** Representative immunofluorescent staining of MARCO+ marginal zone macrophages (blue) and CD3+ (green) T cells that also showed negative staining for CD4, CD8, and NK1.1 (red) in the spleens from the indicated mice. Upper: magnification, ×20. Scale bar: 50 μm; Lower: digitally magnified views of the boxed areas in the upper panels. **f** Flow cytometry quantitation of infiltrating T cells in kidneys of indicated mice. **g** Bar graphs show absolute cell numbers of indicated cell population of kidneys from mice with indicated treatment. Data represent the mean ± SEM, *$P < 0.05$, **$P < 0.01$, **$P < 0.005$ vs. indicated control, two-tailed Student's $t$ test.

above. Of note, those newly formed DN T cells gained the capacity to produce IL-17 (Fig. 3b).

To further verify our findings in different murine models, another mixed cell transfer experiment was carried out in which flow cytometry sorted CD8 T cells (TCRβ+NK1.1−CD8+) from normal *Cd45.2* B6.*DsRed* mice were transferred along with CD8 T cells from *Cd45.2* B6 *Aire*−/− mice[26] (Fig. 3b) into *Cd45.1* B6 recipients. The response of transferred T cells to co-transferred apoptotic thymocytes and the possible downregulation of CD8 were assessed by flow cytometry. Consistently, significantly higher numbers of transferred T cells from *Cd45.2* B6 *Aire*−/− were observed in the spleens of the recipients receiving CLs plus apoptotic B6 thymocytes than the spleens of matched controls or those receiving only apoptotic B6 thymocytes (Fig. 3c; Supplementary Fig. 6b). As expected, the more the cells proliferate the higher the CD8 loss and IL-17 production occur (Fig. 3c, d). Together, we provide compelling evidence proving the feasibility of conversion of CD8 T cells into DN T cells in lupus-prone mice, which contribute to lupus pathogenesis.

**IL-23 favors DN T-cell formation**. Absence of MZMs probably builds up an inflammatory milieu which mimics disturbed

cytokine production in lupus[18]. We hypothesized that the skewed serum cytokine profile might favor the conversion of CD8 T cells into DN T cells upon antigen stimulations by providing secondary signals. To test this possibility, we measured serum cytokine levels after the administration of apoptotic cells (ACs) in the presence or absence of MZMs. It is well described that TGF-ββ1 is essential for immune-tolerance induction[27], but IL-6 and IL-23 contribute to the erosion of tolerance against self[28]. We found this to be the case because AC injection in control mice led to a large increase in serum TGF-β1 levels. In contrast, MZM depletion reduced TGF-β1 induction after AC injection coupled with increased IL-23 and IL-6 levels (Fig. 4a). To determine the possible direct effects of IL-23 on DN T-cell generation, LPS or β-1,3-glucan (Curdlan, recognized by the membrane-bound Dectin-1 receptor leading to IL-23 production)[29,30] was injected i.p. into mice prior to the co-transfer of OT-I T cells and apoptotic m-OVA+ thymocytes. Both LPS and Curdlan promoted a cytokine storm but only Curdlan promoted IL-23 production with simultaneously downregulation of IL-12[29,30]. Flow cytometry revealed that a significantly higher percentage of transferred OT-I T cells exhibited a reduction of CD8 expression in the spleens of mice which received Curdlan compared with other groups (Fig. 4b).

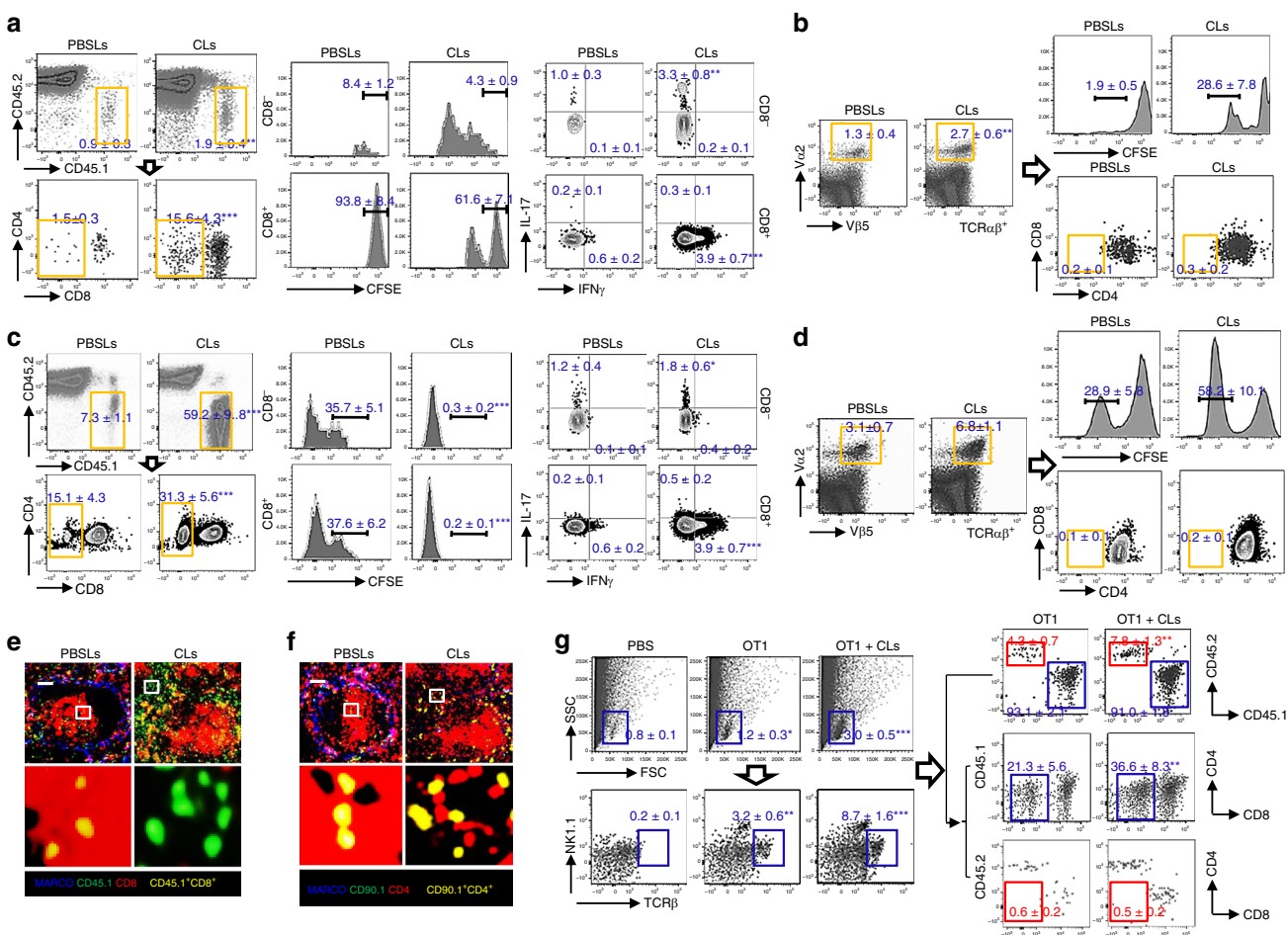

**Fig. 2 Exposure to apoptotic cell debris induces loss of CD8, but not CD4 expression.** CD8 T cells from *Cd45.1* OT-I TCR Tg *Rag1*−/− B6 mice (**a**) or CD4 T cells from *Cd90.1* OT-I TCR Tg *Rag1*−/− B6 mice (**b**) were labeled with CFSE and i.v. transferred. Recipients were administered with CLs or control liposomes, apoptotic m-OVA Tg thymocytes, 12 h and 16 h later sequentially. Mice were sacrificed after an additional 72 h. **a** Left: flow cytometry analysis of cell surface CD8 expression on transferred OT-I CD8 T cells; middle: flow cytometry analysis of in vivo OVA-antigen-specific responses indicated by attenuation of CFSE intensity. Right: flow cytometry quantitation of the percentage of IL-17+ and IFN-γ+ cells in CD8− and CD8+ populations. **b** Flow cytometry analysis of cell surface CD4 expression on transferred OT-II Vα2+Vβ5+ CD4 T cells. In vivo OVA responses were indicated by attenuation of CFSE intensity. CD8 from *Cd45.1* OT-I *Rag1*−/− B6 mice (**c**, **e**, **g**) or CD4 T cells from *Cd90.1* OT-II *Rag1*−/− B6 mice (**d**, **f**) were labeled with CFSE and i.v. transferred into m-OVA Tg B6 mice administered with CLs or control liposome 4 h before and sacrificed after an additional 72 h. **c** Left: Flow cytometry analysis of cell surface CD8 expression on transferred OT-I T cells. Middle: flow cytometry analysis of in vivo OVA responses indicated by attenuation of CFSE intensity. Right: flow cytometry quantitation of the percentage of IL-17+ and IFN-γ+ cells in CD8− and CD8+ populations. **d** Flow cytometry analysis of cell surface CD4 expression on transferred OT-II T cells. In vivo OVA responses were indicated by attenuation of CFSE intensity. **e**, **f** Immunofluorescent staining of transferred OT-II (**e**) or OT-I T cells (**f**). Upper: magnification, ×20. Scale bar: 50 μm; Lower: digitally magnified views of the boxed areas in the upper panels. **g** Flow cytometry analysis of infiltrating T cells from both donors and recipients in the kidneys from indicated mice. Data represent the mean ± SEM, *P < 0.05, **P < 0.01, ***P < 0.005 vs. indicated control, two-tailed Student's t test. n = 4–5 mice per group for two independent experiments.

To further validate the role of IL-23 on DN T cells in lupus-prone mice, we overexpressed IL-23 in wild-type B6.*lpr* mice at 8 weeks of age by hydrodynamic administration of a minicircle (MC) cDNA construct encoding *Il23p19* and *Il12/23p40* (IL-23 MC)[31] (Supplementary Fig. 7). We observed that in vivo expression of IL-23 alone was sufficient to drive the expansion of DN T cells without affecting the integrity of MZM barrier (Supplementary Figs. 8 and 9), which is consistent with our previous reports that *Il23r*-deficient lupus-prone B6.*lpr* mice have decreased numbers of DN T cells[32]. In accordance with the enlargement of DN T-cell pool, the mice with IL-23 MC administration showed enhanced autoimmunity, including elevated serum anti-dsDNA titers, earlier appearance of proteinuria, increased intrarenal lymphocyte infiltration and immune complex deposition and etc. (Fig. 4c–f; Supplementary Fig. 10). Notably, enhanced IL-17 production in DN T cells

from both spleens and kidneys was observed (Supplementary Figa. 11 and 12). Thus, persistent stimulation with IL-23 in vivo promotes the development of autoimmunity by enhancing DN T-cell generation. The differentiation and expansion of DN T cells from polyclonal CD8 T cells of autoimmune mice after exposure to exogenous ACs indicate at least a great portion of them are self-reactive. Antigen-specific T cells are implicated in SLE. U1-70, U1-A, and U1-C, together with U1-RNA and the seven Smith proteins, compose the U1-small nuclear ribonucleoprotein (U1-snRNP) complex. The frequency of Th17 cells specific for U1-70-snRNP has been shown to correlate with U1-70 autoantibody production and disease severity[33]. Many shared characteristics between IL-17+ DN T cells with Th17s bring the question whether there is a portion of IL-17+ DN T cells which also recognize U1-70. To address this, we generated IL-17 GFP B6.*lpr* mice and administered them IL-23

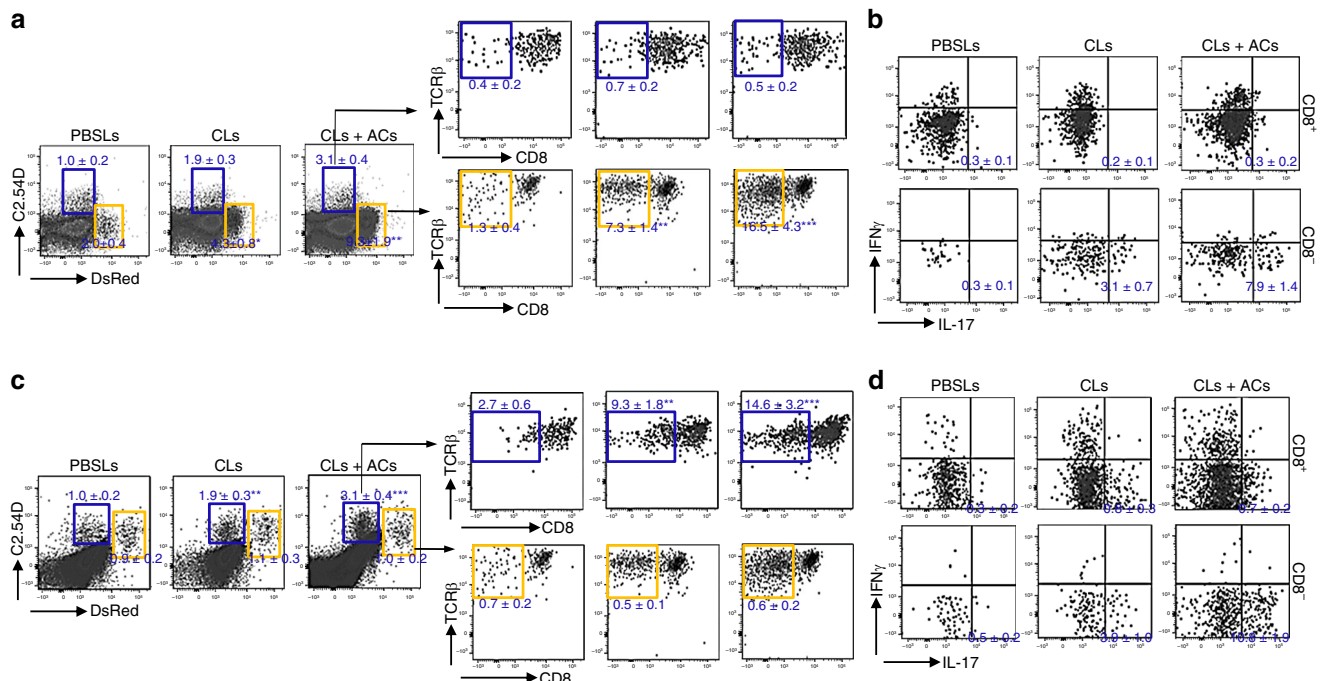

**Fig. 3 Autoreactive Polyclonal CD8 T cells differentiate into DN T cells.** A mixed population of purified CD8 T cells (TCRβ+NK1.1−CD8+, 2 × 10⁶/mouse, 1:1 ratio) from *Cd45.1* B6 mice and *Cd45.2* B6.*lpr.DsRed* (**a**, **b**) or from normal *Cd45.2* B6.*DsRed* mice and *Cd45.2* B6 *Aire*−/− mice (**c**, **d**) were transferred into either *Cd45.2* B6 (**a**, **b**) or *Cd45.1* B6 (**c**, **d**) recipients. Twelve hours later, recipients were administered with control liposome, CLs, CLs plus apoptotic thymocytes prepared from B6 mice. Mice were euthanized after an additional 72 h. *n* = 5 mice per group for two independent experiments. **a** Left: flow cytometry analysis of the percentage of the transferred CD8 T cells in the spleens of *Cd45.2* B6 recipients with the indicated treatment. Right: flow cytometry analysis of CD8 expression on the surface of the transferred CD8 T cells in the spleens of *Cd45.2* B6 recipients with indicated treatment. **b** Flow cytometry quantitation of the percentage of IL-17+ and IFN-γ+ cells in CD8− (upper) and CD8+ (lower) populations derived from transferred CD8 T cells. **c** Left: flow cytometry analysis of the percentage of the transferred CD8 T cells in the spleens of *Cd45.1* B6 recipients with indicated treatment. Right: flow cytometry analysis of CD8 expression on the surface of the transferred CD8 T cells in the spleens of *Cd45.1* B6 recipients with indicated treatment. **d** Flow cytometry quantitation of the percentage of IL-17+ and IFN-γ+ cells in CD8− (upper) and CD8+ (lower) populations derived from transferred CD8 T cells. Data represent the mean ± SEM (*P < 0.05, **P < 0.01, ***P < 0.005 vs. PBSLs-treated control, two-tailed Student's *t* test.

MC. Three months later, GFP expression was observed in both CD4 T and DN T cells. Next, we purified splenic T cells and co-cultured them with irradiated wild-type B6 spleenocytes loaded with U1-70 peptide in the presence of brefeldin A. Production of IL-17 but not IFNγ was detected in GFP+ CD4 T cells and GFP+ DN T cells (Fig. 4g), which indicated that a portion of IL-17+ DN T cells could respond to self-antigen U1-70. Furthermore, U1-70:I−Aᵇ tetramers were applied to identify IL-17-producing T cells, which recognize U1-70 in the context of MHC class II. Consistent with the fact that DN T cells derive from self-reactive CD8 but not CD4 T cells, U1-70:I-Aᵇ tetramer-positive T cells were only found in CD4+, but not CD8+ or DN T-cell lineages (Fig. 4h). Collectively, in lupus-prone mice, U1-70–specific DN T cells respond to U1-70 and produce IL-17A in MHC class I, but not class II-dependent manner.

DN T cells from SLE patients have been shown to provide help to B cell for antibody production in vitro[34]. To validate the help from DN T cells to B cells for autoantibody production in vivo, splenic DN T cells were enriched from IL-23 MC-treated B6.*lpr* mice and transferred into B6 *Rag1*−/− mice which had received purified B cells from 12-months-old B6.*lpr* spleens 1 day earlier. The same recipients which have been injected with PBS, splenic CD8 or CD4 T cells were used as controls. Autoantibodies in the circulation were examined 1 month after cell transfer. The group which had received DN T cells and B cells displayed higher titers of circulating autoantibodies and more renal immune complex deposition compared with groups which had received B cells

alone or B cells and CD8 T cells (Supplementary Fig. 13). This finding provides first evidence that DN T cells contribute to lupus pathogenesis in vivo by promoting autoantibody production and renal immune complex deposition.

**Myeloid *Tgfb1* deficiency promotes DN T-cell formation.** Next, we determined the possible suppressive effect of the tolerogenic cytokine TGF-β1 released from MZMs. CFSE-labeled OT-I or OT-II T cells were transferred into WT, *Lyz2*cre+*Tgfb1*fl/fl, or *Lyz2*cre+*TgfbrII*fl/fl (target gene deletion in myeloid cell lineage), mice and the response of transferred cells to apoptotic m-OVA+ thymocytes was assessed by flow cytometry. The "leaking" pro-liferation of either OT-I or OT-II T cells was only observed in *Lyz2*cre+*Tgfb1*fl/fl, but not WT or *Lyz2*cre+*TgfbrII*fl/fl recipients (Fig. 5a). Consequently, the reduced expression of CD8 was observed on proliferating OT-I but not OT-II T cells (Fig. 5b). Since both macrophages and dendritic cells (DCs) have been proposed to sustain immune tolerance to apoptotic cells by virtue of producing anti-inflammatory cytokine TGFβ[35,36], we asked whether MZMs were the main cellular sources of TGFβ during the clearance of circulating apoptotic cells. Apoptotic thymocytes were administered into mice pre-treated with or without CL to deplete MZMs. Thirty minutes later, different population myeloid cells including MZMs (CD11bloF4/80−SIGN-R1+MHCII−), red pulp macrophages (RPMs CD11bloF4/80+CD68+), CD4+ DCs (CD11b+CD4+), and CD8+ DCs (CD11b+CD8+) were FACS sorted, and the production of various cytokines including *Tgfb1* was examined by qRT-PCR. As expected, when the marginal zone

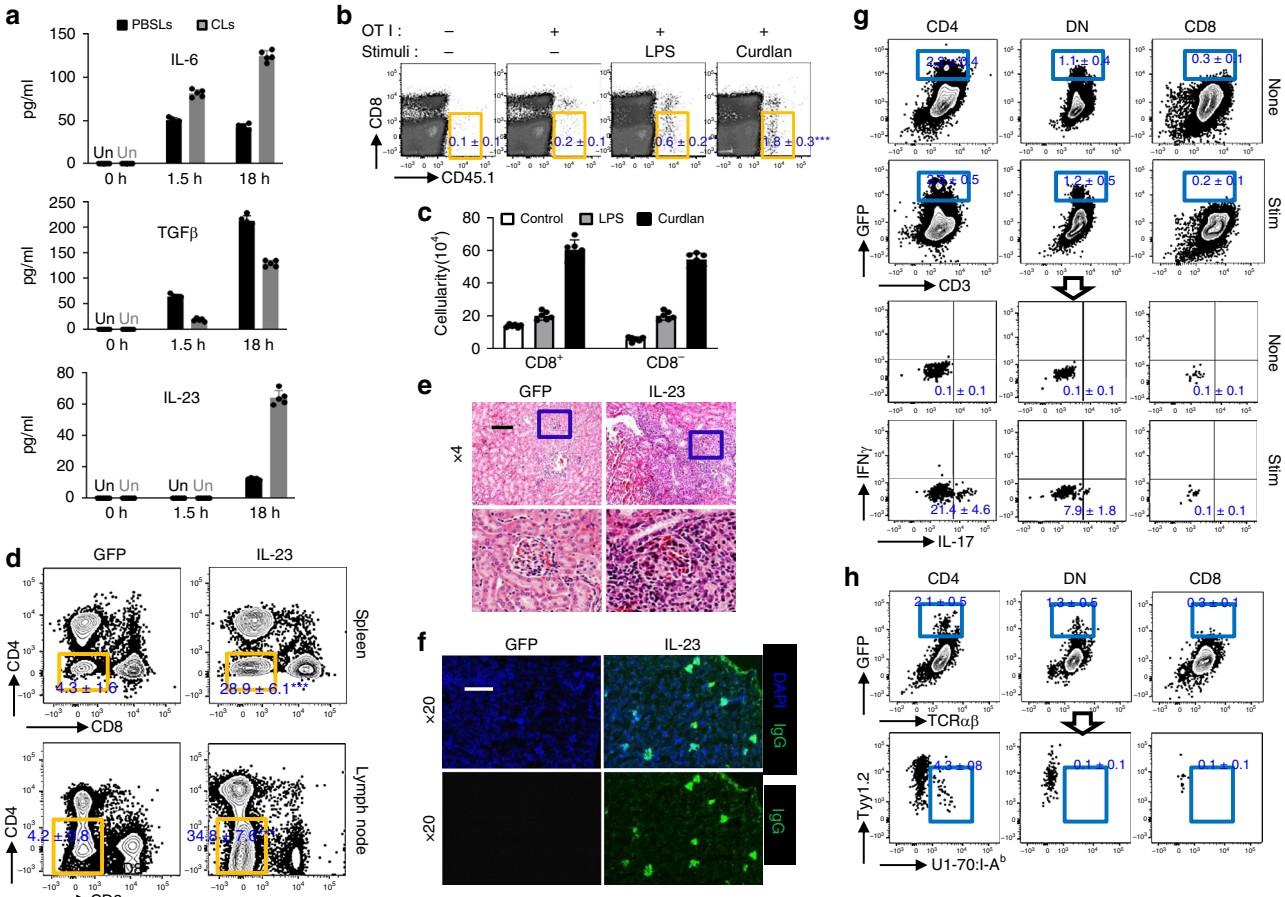

**Fig. 4 IL-23 favors DN T-cell formation.** B6 mice were i.p. injected with apoptotic thymocytes prepared from B6 mice 4 h after control liposome or CLs i.v. administration, and blood was collected after indicated time. **a** ELISA analysis of indicated cytokines in the serum from indicated mice after indicated time, $n = 5$ per group. **b, c** CD8 T cells from *Cd45.1* OT-I TCR Tg *Rag1*$^{-/-}$ B6 mice were i.v. transferred into B6 mice. Twelve hours later, recipients were administered with apoptotic thymocytes prepared from m-OVA Tg B6 mice plus PBS, LPS (2 mg/kg) and Curdlan (5 mg/kg) separately. Mice were euthanized after an additional 72 h. $n = 6$ per group. **b** Flow cytometry analysis of CD8 expression on transferred OT-I CD8 T cells from spleens of indicated recipients. **c** Bar graphs show quantitation of the absolute cell numbers of both CD8$^+$ and CD8$^-$ CD45.1$^+$OT-I T cells from the spleens of indicated recipients. **d** Flow cytometry quantitation of the percentage of CD3$^+$TCRb$^+$CD4$^-$CD8$^-$ DN T cells in spleens or lymph nodes from mice with the indicated MC administration. **e** Representative H&E images of kidneys from mice with indicated MC administration. Upper: magnification, ×4, scale bar: 200 μm; Lower: digitally magnified views of the boxed areas in the upper panels. **f** Fluorescence microscopic images of IgG staining in the kidneys of mice with indicated MC administration. Magnification, ×10, scale bar: 100 μm. **g** IL-23 MC was administered to 2-months-old IL-17 GFP B6.*lpr* mice. Three months after administration, splenic T cells were purified and co-cultured with or without irradiated wild-type B6 splenocytes pre-loaded with U1-70 peptide. Top: flow cytometry quantitation of the percentage of GFP$^+$ cells CD4, CD8 and DN T cells individually. Bottom: flow cytometry analysis of the production of IL-17 and IFNγ in GFP$^+$ populations from indicated T cells. **h** Flow cytometry quantitation of the percentage of U1-70:I-A$^b$ tetramer-positive cells in GFP$^+$ population from indicated T cells. Data represent the mean ± SEM (*$P < 0.05$, ***$P < 0.005$ vs. control, Student's $t$ test; $n = 4$ mice per group for two independent experiments for (**e–h**).

barrier was intact, MZMs expressed higher TGFβ1 compared with other myeloid cells after exposure to apoptotic cells (Fig. 5c). However, in the absence of MZMs, we observed the activation of other myeloid cells after exogenous AC administration, indicated by the elevation of pro-inflammatory cytokines, including IL-6 and IL-23 especially by RPMs (Fig. 5d). Of note, in contrast to RPMs and CD4$^+$ dendritic cells, which produce pro-inflammatory cytokines, CD8$^+$ dendritic cells produce high levels of the immune suppressive cytokine TGFβ1 only when MZM barriers were disrupted (Fig. 5d)[37,38]. To track the apoptotic cell clearance in vivo, CFSE-labeled ACs were administered into mice with or without CL pre-treatment. Confocal images revealed that the numbers of CFSE$^+$ cells, which were located predominately in the MZ, peaked at 5 min, and the majority was phagocytosed at 20 min (Fig. 5e). However, the leakage of apoptotic cell debris to follicle resident DCs, for example CD8$^+$

dendritic cells, was only observed in the absence of MZMs (Fig. 5f). Of note, RPMs were well preserved after CL treatment, and follicular translocation of these cells after AC administration in the absence of MZMs suggested follicular Ag transportation by RPMs may drive the activation of T and B cells against self-antigens (Fig. 5e). Flow cytometry analysis further confirmed our observation. Collectively, our results strongly suggest that under naive status, MZMs are the main cellular source of TGFβ1 to maintain immune tolerance to apoptotic cells.

To assess whether *Tgfb1* deficiency in macrophages promotes DN T-cell formation in our findings in lupus-prone mice, we backcrossed *Lyz2*$^{cre+}$*Tgfb1*$^{fl/fl}$ mice to B6.*lpr* mice. Enlarged spleens and lymph nodes with significantly increased cell counts were observed (Fig. 5h; Supplementary Fig. 14). FACS analysis revealed that there were significantly increased percentages of DN T cells in the spleens of *Lyz2*$^{cre+}$*Tgfb1*$^{fl/fl}$ B6.*lpr* mice than in

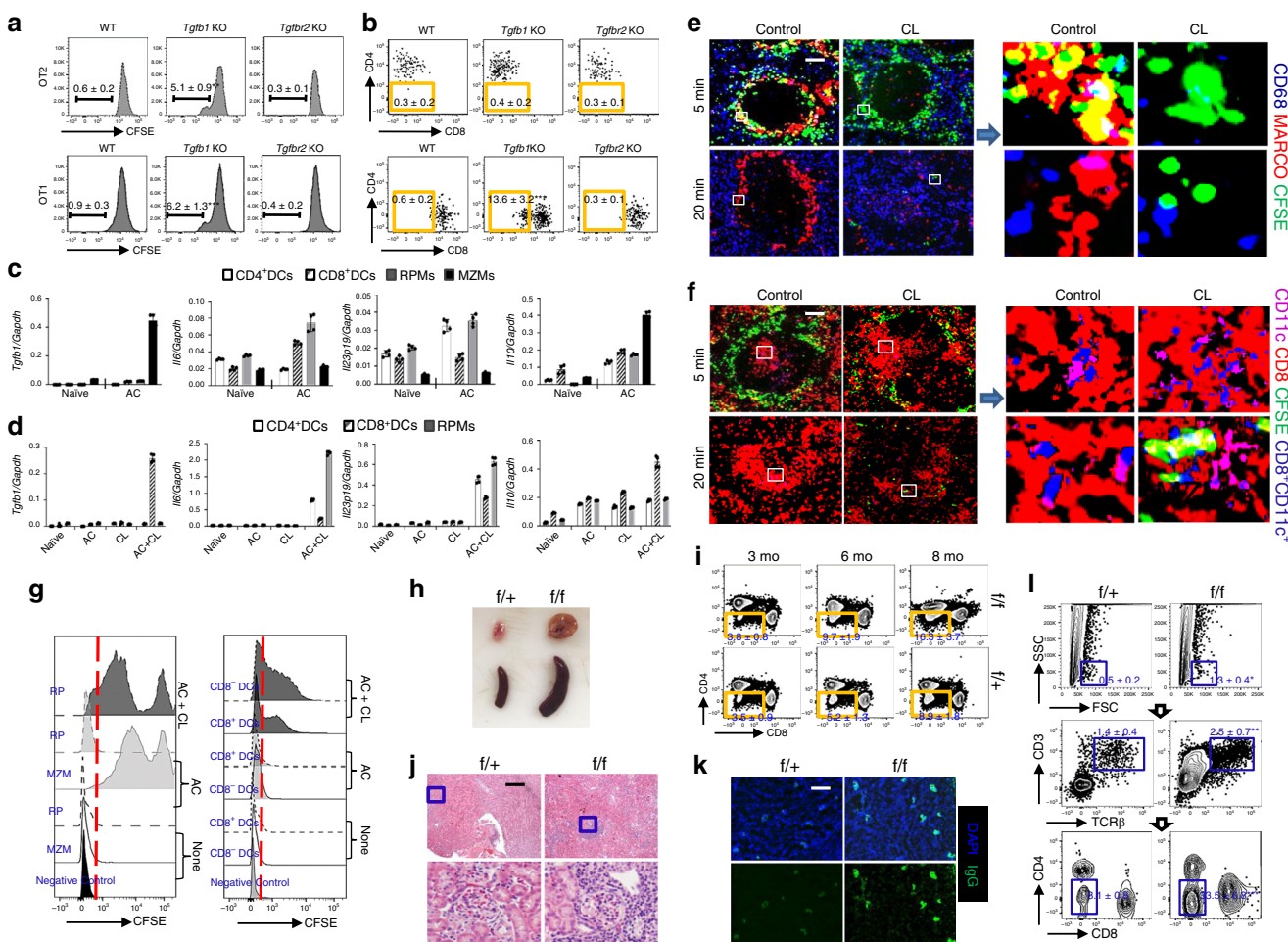

**Fig. 5 Deficiency of *Tgfb1* in macrophages promotes DN T-cell formation. a, b** CFSE-labeled CD4 T cells from OT-II *Rag1*−/− or CD8 T cells from OT-I *Rag1*−/− B6 mice were co-transferred with apoptotic m-OVA thymocytes into the indicated recipients. Recipients were euthanized after an additional 72 h. *n* = 4 per group for two independent experiments. Flow cytometry analysis of proliferation (**a**) or surface CD4/CD8 expression (**b**) on transferred OT-I or OT-II T cells. **c, d** Apoptotic thymocytes were administered into mice pre-treated without (**c**) or with CL (**d**) for 2 h. MZMs, red pulp macrophages (RPMs or RPs), CD4+DCs, and CD8+DCs were FACS sorted 30 min later. qRT-PCR analysis of indicated gene expression in indicated cell populations sorted from mice administered with apoptotic thymocytes (**c**) or mice administrated with CLs plus apoptotic thymocytes (**d**). *n* = 4 per group. **e–g** CFSE-labeled apoptotic thymocytes were administered into mice with or without CL pre-treatment. *n* = 4 per group. Magnification, ×20, scale bar: 50 μm. **e** Confocal microscopic images show that in the absence of MZMs, administration of apoptotic thymocytes leads to follicular translocation of RPMs (CD68+). **f** Confocal microscopic images show the uptake of apoptotic cell debris by CD8+ dendritic cells in the absence of MZMs. **g** Flow cytometry shows the apoptotic cell uptake (CFSE+) by indicated populations with indicated treatment. **h, j–l** Mice were at 12 months of age. **h–l** *n* = 5 per group for two independent experiments. **h** Representative photos of spleens and lymph nodes from indicated mice. **i** Flow cytometry quantitation of CD3+ TCRβ+CD4−CD8− DN T cells in spleens or lymph nodes from indicated mice at indicated ages. **j** Fluorescence microscopic images of IgG staining in the kidneys of indicated mice. Magnification, ×10, scale bar: 100 μm. **k** Representative H&E images of kidneys from indicated mice. Upper: magnification, ×4, scale bar: 200 μm; Lower: Digitally magnified views of the boxed areas in the upper panels. **l** Flow cytometry analysis of infiltrating T cells in the kidneys from indicated mice. Data represent the mean ± SEM (*P < 0.05, **P < 0.01, ***P < 0.005 vs. control, two-tailed Student's *t* test).

those of wild-type matched controls starting at 6 months of age (Fig. 5i). Consistently, the *Lyz2*cre+*Tgfb1*fl/fl B6.*lpr* mice also displayed other enhanced manifestations of lupus characteristics including higher sera titers of both IL-17 and dsDNA antibodies, earlier appearance of proteinuria, more intrarenal lymphocyte infiltration and IgG-containing immune complex deposition in the glomeruli even though the marginal zone barriers are intact (Fig. 5j, k; Supplementary Figs. 14–18). Thus, TGFβ1 production from MZMs restricts the development of DN T cells in lupus-prone mice which results in reduced autoimmune manifestations.

**DN T cells display proliferating or proliferated phenotype.** If activation-induced CD8 downregulation is the major driver for DN T-cell generation, the newly formed DN T cells should

display proliferating or proliferated phenotype. To confirm this MRL.*lpr* or control MPJ mice (same background without recessive *lpr* gene mutation) were injected i.p. 5-bromo-2′-deoxyuridine (BrdU, an analog of the nucleoside thymidine used to identify proliferating cells) and cell cycle analysis on different T-cell subtypes was performed by immunofluorescent staining of BrdU in conjunction with 7-AAD (a fluorescent compound with a strong affinity for DNA). Significantly higher numbers of DN T cells entered the S phase in the spleens of MRL.*lpr* mice than the spleens of age-matched MPJ mice (Fig. 6a–c). Correspondingly, the percentages of DN T cells positive for BrdU were also higher in MRL.*lpr* mice. Since self-antigen-activated CD8 T cells preferentially differentiate into DN T cells, it is reasonable that DN T cells acquired a more active status. To further assess our

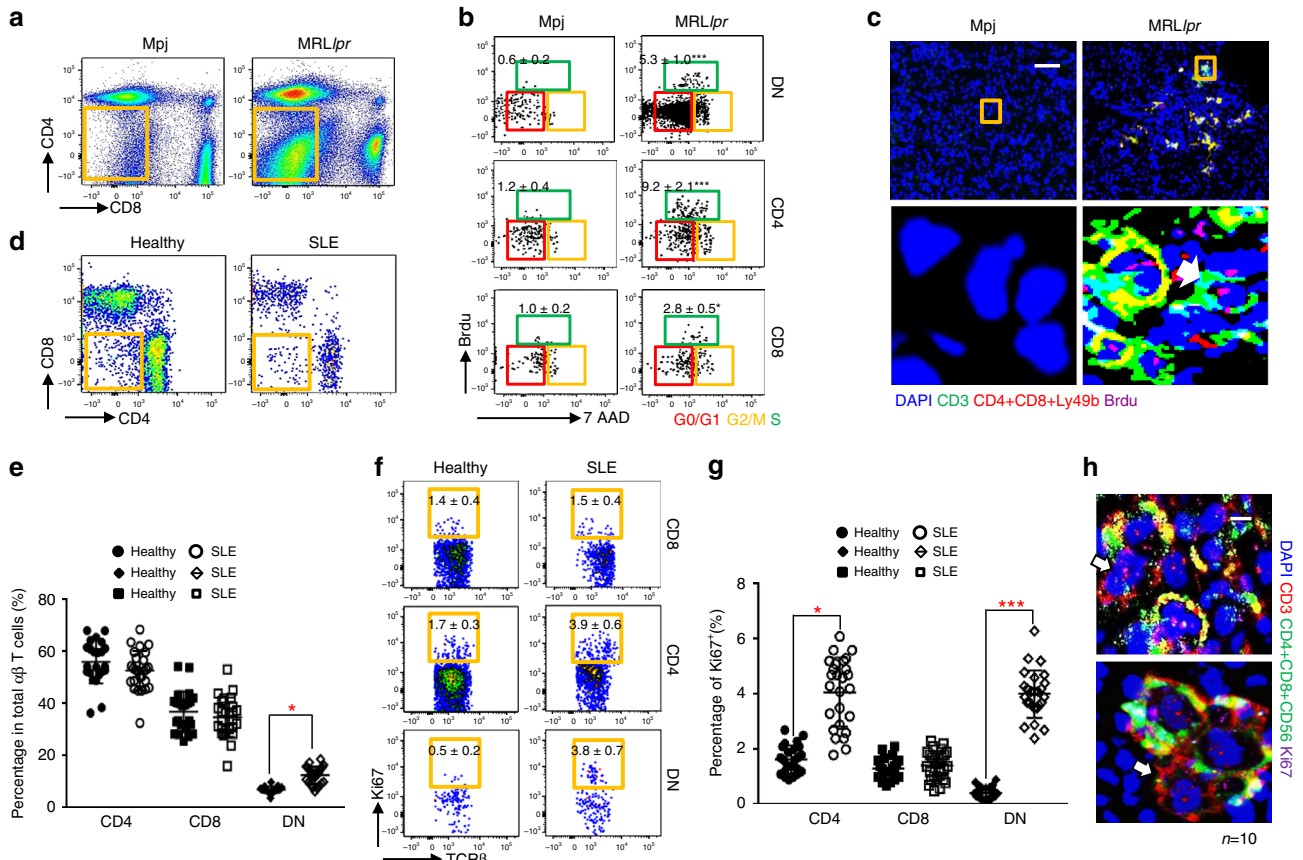

**Fig. 6 DN T cells display proliferating or proliferated phenotype. a–c** Sixteen week-old female MPJ or MRL.*lpr* mice were injected i.p. with BrdU (1 mg/mouse) 12 and 0.5 h prior to the euthanasia. Data represent the mean ± SEM (*$P < 0.05$, ***$P < 0.005$ vs. control, Student's $t$ test; $n = 6$ mice per group). **a** Flow cytometry analysis of different subset T cells from indicated mouse spleens. **b** Flow cytometry analysis of cell cycle phases of different subset T cells from indicated mouse spleens (gated in CD3+TCRβ+Ly49b−). BrdU incorporation is used for characterization of cells that are actively synthesizing DNA and 7-AAD staining intensities were further applied to define related cell phase in the cell cycle (G0/G1, G2/M, or S phases). **c** Representative immunofluorescent staining of BrdU+ (magenta) DN T cells (green) in kidneys from indicated mice. Upper: magnification, ×10. Scale bar: 100 µm; Lower: digitally magnified views of the boxed areas in the upper panels. **d–g** $n = 23$ for healthy controls and $n = 26$ for SLE patients. **d** Representative flow cytometry analysis of CD4, CD8, and DN T cells in total T cells in PBMC from either healthy subjects or SLE patients. **e** Scatter plots show the percentage of CD4, CD8, and DN T cells in total circulating T cells (gated in CD3+TCRβ+CD56−) from either healthy controls or SLE patients. **f** Flow cytometry analysis of Ki67+ CD4, CD8, and DN T cells in total T cells from PBMC from healthy controls or SLE patients. **g** Scatter plots show the quantitation of Ki67+ CD4, CD8, and DN T cells of human PBMC (gated in CD3+TCRvβ+CD56−) from either healthy controls or SLE patients. **h** Representative immunofluorescent staining of Ki67+ (magenta) DN T cells (green) in kidney biopsy samples from patients with SLE. Upper: magnification, ×100. Scale bar: 5 µm; Lower: digitally magnified views of the boxed areas in the upper panels. $n = 10$. Data represent the mean ± SEM (*$P < 0.05$, ***$P < 0.005$ vs. indicated control, Student's $t$ test).

hypothesis in SLE patients, we examined the Ki67 expression in circulating T cells from both healthy and SLE patients. We found this to be the case since flow analysis revealed a significant increase of Ki67+ DN T cells in SLE patients (CD3+TCRβ+CD56−CD4−CD8−) compared with healthy controls (Fig. 6d–h). Conversely, no differences in the percentage of Ki67+ CD8 T cells were observed between SLE patients and healthy controls (Fig. 6f, g).

**Skewed TCR Vβ repertoires and diluted TREC of DN T cells.** Antigen-driven selection of T cells can lead to a narrowed TCR Vβ repertoire[39]. To confirm this, we assessed the Vβ gene usage of CD4, CD8 and DN T cells from both B6 and B6.*lpr* mice at the indicated ages. Diverse Vβ gene usage by different subset T cells from the indicated mice was revealed by flow cytometry. Interestingly, DN T cells in spleens from 10-months-old B6.*lpr* mice displayed increased frequency of TCR Vβ chain encoded by Vβ5.1/5.2, Vβ8.1/8.2, and Vβ8.3 gene families compared with either 10-month-old B6 or 5-month-old B6.*lpr* mice (Fig. 7a).

Importantly, marginal zone macrophage depletion accelerated this age-associated enrichment, suggesting that the dominance of Vβ5 and Vβ8 might be a consequence of selective expansion driven by self-antigens (Fig. 7a). Of note, T-cell receptor (TCR) beta chain sequence analysis revealed a higher proportion of top ten clonotypes in DN T cells from MRL.lpr mouse which confirmed the selective expansion of some clone types of DN T cells (Supplementary Table 1, Supplementary Fig. 19). Consistently, increased clonal space occupation by Vβ5 clonotypes was also observed in DN T cells from MRL.lpr mouse (Supplementary Fig. 20, Supplementary Table 2). We next utilized human peripheral blood mononuclear cells (PBMCs) isolated from SLE patients and healthy donors, and confirmed the preferential usage of Vβ5 and Vβ8 by CD8 and DN T cells from SLE patients but not healthy donors (Fig. 7b), further suggesting the possibility that DN T cells develop from self-antigen-stimulated CD8 T cells in SLE.

Peripheral proliferation could also be determined by T-cell receptor excision circle (TREC) dilution analysis since TRECs are

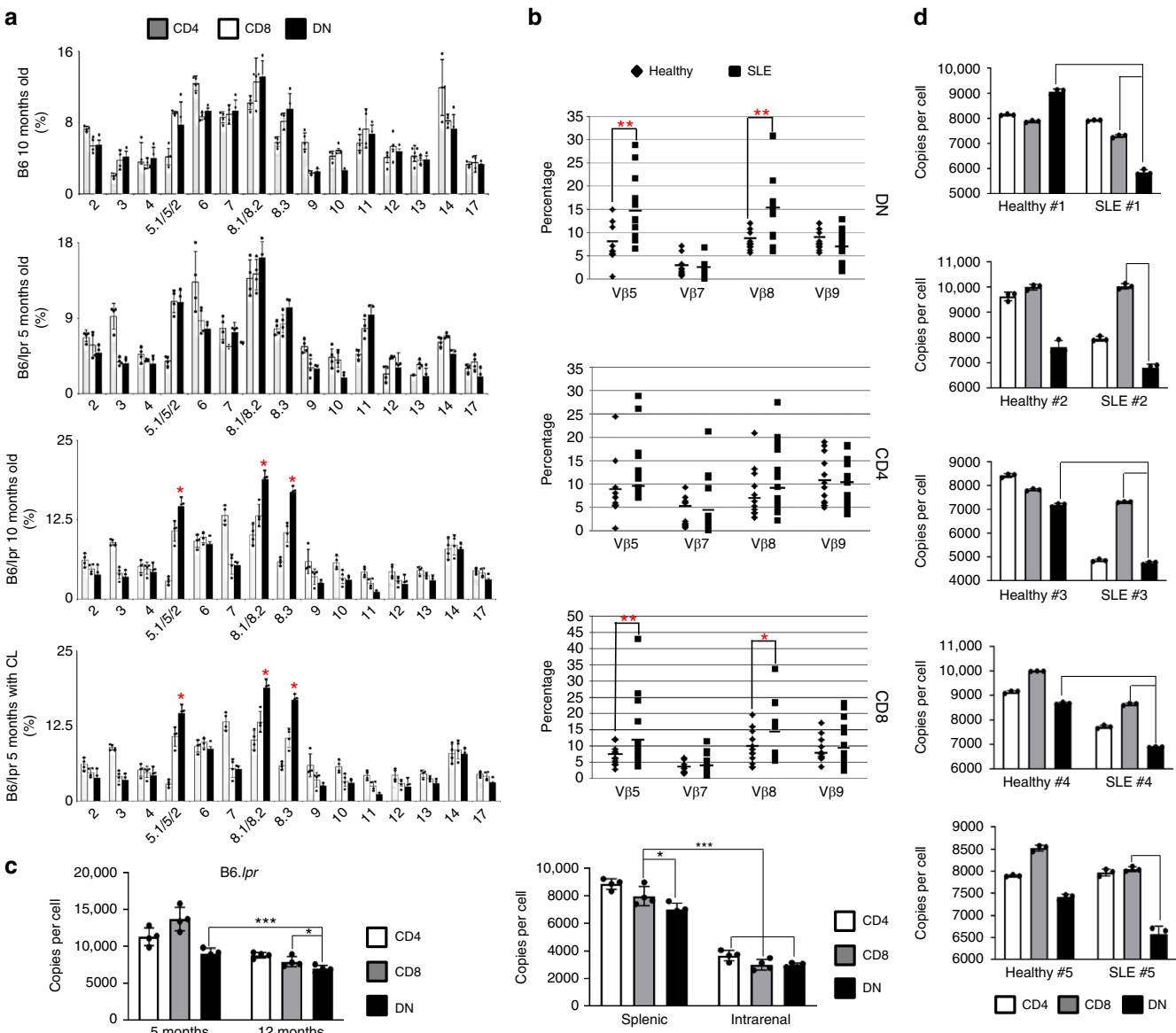

**Fig. 7 Skewed TCR Vβ repertoires and diluted TREC of DN T cells. a** Bar graphs show quantitation of TCR Vβ usage by CD4, CD8, and DN T cells of spleens from indicated mice. CD4 T cells: CD3+TCRβ+NK1.1−CD4+, CD8 T cells: CD3+TCRβ+NK1.1−CD8+, DN T cells: CD3+TCRβ+NK1.1−CD4−CD8−, bottom: female B6.lpr mice were treated with clodronate liposomes (CLs, 100 ug/mouse) every other week for 2 months total starting at 3 months of age. n = 4 mice per group. **b** Scatter plots show the quantitation of TCR Vβ usage by DN, CD4, and CD8 T cells in PBMC from either healthy controls or SLE patients. CD4 T cells: CD3+TCRβ+CD56−CD4+, CD8 T cells: CD3+TCRβ+CD56−CD8+, DN T cells: CD3+TCRβ+CD56−CD4−CD8−. n = 12 each group. **c** Left: quantitative real-time PCR analysis of TCR excision circles in CD4, CD8, and DN T cells in spleens from B6.lpr with indicated ages. Right: quantitative real-time PCR analysis of TCR excision circles in CD4, CD8, and DN T cells from either spleens or kidneys of 12-months-old female B6.lpr mice (n = 4 mice per group). **d** Quantitative real-time PCR analysis of TCR excision circles in CD4, CD8, and DN T cells (gated in CD3+TCRαβ+CD56−) in PBMC from five SLE patients and five matched healthy controls with three technical replicates for each sample. Data represent the mean ± SEM (*P < 0.01, **P < 0.05, ***P < 0.005 vs. indicated controls). Two-tailed unpaired Student's t test was used when only two groups were compared, and the one-way ANOVA was applied with comparison of more than two groups.

extrachromosomal DNA byproducts of T-cell receptor (TCR) rearrangement[40], which are non-replicative. Quantitative real-time PCR assay revealed that the TREC numbers in all T-cell subtypes from MRL.lpr mice were reduced as compared with cells from MPJ mice (Fig. 7c). Moreover, a significant reduction of TREC numbers was observed in all subsets of intrarenal T cells compared with their counterparts from spleens (Fig. 7c). Importantly, the DN T cells have the lowest TREC numbers, suggesting that the DN T cells acquire a proliferated phenotype in vivo (Fig. 7c). To build a clinical connection with human SLE, we measured TRECs in sorted T cells from both SLE patients and

healthy controls. Since TRECs generally show an inverse correlation with age, comparison was only conducted between indicated SLE patients with strictly age-matched healthy controls. Significant reductions of TREC numbers in DN T cells from SLE patients were observed in all compared groups (Fig. 7d), suggesting they went through clonal proliferation and expansion in vivo.

## Discussion

The importance of expanded DN T cells has been exploited in several autoimmune diseases, including SLE[34], autoimmune

lymphoproliferative syndrome[41], Sjogren's syndrome[42], and aplastic anemia[43], which are believed to be driven/exaggerated by autoantigens derived from uncleared apoptotic cell blebs[2,44]. MZMs surrounding the splenic follicles have been reported with the capability of both efficient clearance to ACs and induction of immune tolerance to ACs[15,17,18]. Numerical reduction and functional defects of marginal zone macrophages were shown not only in lupus-prone mice but also in SLE patients[16,18–20], which suggest that they are common features for lupus. Here, we provided the missing link between the impaired apoptotic cell clearance with DN T-cell expansion in vivo (Supplementary Fig. 21). First, we demonstrated self-antigens derived from apoptotic cells can activate self-reactive CD8 T cells, which give rise to DN T cells through the downregulation of CD8 expression on the cell surface. CD8 T cells specific for antigen delivered to immune-privileged sites were described as CD8 $T_{regs}$ by limiting effector T-cell responses[45–49]. The importance of CD8 expression for CD8 T-cell mediated suppression was shown by BXSB. In $Yaa$ $Cd8\alpha^{-/-}$ mice, the deficiency of CD8$^+$ suppressor T cells leads to an accelerated and more severe form of spontaneous lupus-like autoimmune syndrome compared with BXSB.$Yaa$ inbred mice[50]. The whole process of conversion from CD8 into DN T cells contributes to the pathogenesis of lupus considering the fact that this self-reactive T population lost their potential regulatory function and acquired a pro-inflammatory phenotype, including enhanced tissue migration ability and IL-17 production. Our study further underscores the contribution of DN T cells in the pathogenesis of autoimmunity by providing help to B cells to produce autoantibodies. This finding is consistent with the observation that the therapeutic effects of rapamycin in SLE were associated with the reduction of IL-17-producing DN T cells[51]. Of note, rapamycin did not block the polarization of anti-inflammatory macrophages in vitro[52], which suggested the intrinsic requirement of mTOR for DN T-cell activation.

Since CD8 expression is essential in stabilizing the interaction between TCR with antigens presented on MHC I complex, it remains unclear what factors help maintain the pathogenic DN T-cell population in vivo. The augmented expansion of DN T cells under inflammatory milieu with MZM deficiency, mimicking the lupus milieu, suggest that skewed combination of pro- and anti-inflammatory cytokine stimulation provides additional signals for DN T-cell activation, expansion, and survival. Of note, our previous report suggests that antigens derived from exogenous infections could not induce downregulation of CD8 expression regardless of the levels of activation or proliferation of antigen-specific CD8 T cells, which is consistent with fact that the expansion of DN T cells was only observed in autoimmune diseases but not in infectious diseases[13]. Last but not least, the clinical observation that DN T cells from SLE patients acquired proliferating or proliferated phenotype (Ki67 expression, diluted TREC, and narrowed TCR repertoire), is concomitant to our findings in mice that autoantigen activated CD8 T cells differentiate into DN T cells.

Potential limitations of our study include the difficulties to assess MZM defects in SLE patients using peripheral blood. Because the circulating monocytes can function as MZM precursors[53], additional studies of human SLE are warranted to explore whether the central defects underlying the gradual changes in MZMs start with circulating precursors. Such information may provide useful tools for early disease diagnoses. Moreover, more evidence is required to confirm that in patients with SLE, abnormalities in marginal zone barrier occur prior to the expansion of DN T cells. Although the presence of DN T cells is not in dispute, they are largely ignored by immunologists because of their poorly understood origin[54]. Here, we provide evidence that disturbed cytokine milieu plus excessive apoptotic debris in lupus facilitate the conversion of self-reactive CD8 T cells to IL-17-producing DN T cells.

## Methods

**Mice**. *DsRed* B6, *OT1* B6, *Cd45.1* B6, *OT-II* B6, *Thy1.1* B6, m-OVA B6, *Rag1*$^{-/-}$ B6, *Il17a-GFP*, *Aire*$^{-/-}$ B6, B6.*lpr*, MPJ, MRL.*lpr* mice were obtained from The Jackson Laboratories. Two-month-old female mice were used unless the age and sex were otherwise specified. All mice were maintained under specific pathogen-free conditions, and all procedures were approved by the Institutional Animal Care Committee of Beth Israel Deaconess Medical Center, Harvard Medical School.

**Human samples**. Peripheral blood from patients with SLE and healthy volunteers and kidney biopsy samples used in this study were procured under Beth Israel Deaconess Medical Center IRB number 2006P000298 and approved informed consent. Kidney tissue from lupus patients was snap-frozen immediately after renal biopsy and stored at −80° until analyzed.

**In vivo AC and CL administration**. For in vivo administration of ACs, mice were injected i.v. with $2 \times 10^7$ apoptotic thymocytes. For depletion of MZMs, mice were treated i.v. with 100 μg of CL (Encapsula NanoSciences, Nashville, TN) 4 h before AC transfer[15]. Control mice were treated with PBS liposomes.

**In vivo cell transfer**. For in vivo cell adoptive transfer, indicated T cells were labeled with CFDA cell tracer (Catalog number: v12883, Thermofisher Scientific) according to manufacture's instruction and transferred i.v. into the indicated recipients ($5 \times 10^6$ per mouse).

**TREC assay**. DNA was purified from T cells using the QIAamp DNA mini kit (Cat No. 51304, QIAGEN). To detect TREC, a real-time quantitative PCR (RQ-PCR) method was used. The primer and probe sequences are listed; human: forward, 5′-CACATCCCTTTCAACCATGCT-3′. Reverse, 5′-GCCAGCTGCA GGGTTTA GG-3′, probe FAM-5′-ACACCTCTGG TTTTTGTAAAGGTGCCCACT-3′-TAMRA. Mouse: forward, FAM-5′-CATTGCCTTTGAACCAAGCTG-3′, reverse, 5′-TTATGCACAGGGTGCAGGTG-3′, probe, FAM–CAG GGC AGG TTT TTG TAA AGG TGC TCA CTT-3′-TAMRA.

**qRT-PCR analysis**. Real-time quantitative PCR was carried out using the following primers: *Il6*, 5′-TCCAGTTTGGTAGCATCCATC-3′ (forward), 5′-CCGGAG AGGAGACTTCACAG-3′ (reverse); *Il10*, 5′-AATTCCCTGGGTGAGAAG-3′ (forward), 5′-TGCAGTTGATGATGAAGATGTC-3′ (reverse); and *Tgfβ1*: 5′-CTACTATGCTAAAGAGGTCAC-3′ (forward), 5′-CATGTTGCTCCACACTTG-3′ (reverse). *Il23*: 5′-AGTGTGAAGATGGTTGTGAC-3′ (forward), 5′-CTGGAG GAGTTGGCTGAG-3′(reverse).

**Histology of frozen sections and confocal imaging analysis**. Mouse spleens or kidneys were embedded in OCT tissue media (Tissue-Tek) and frozen on dry ice. Frozen sections (7-μm thickness) were fixed to slides in ice-cold acetone for 15 min and air dried for 30 s. The sections were blocked with 2% BSA for 30 min at room temperature and then stained for 30 min at room temperature in a humidified chamber with fluorescently labeled antibody cocktails and Hoechst 33258 (Life Technologies). The following fluorescence antibodies were applied according to the manufacturer's instructions: anti-mouse MARCO (BIO-RAD), anti-mouse CD45.1 (Clone A20,Biolegend), anti-mouse CD4 (Clone GK1.5, Biolegend), anti-mouse CD8 (Clone 53–6.7, Biolegend), anti-mouse CD90.1 (Clone OX-7, Biolegend), anti-mouse TCRβ (Clone B20.6, Biolegend), goat anti-mouse IgG cross-adsorbed secondary antibody (ThermoFisher Scientific). All tissue sections were mounted in Prolong Gold Antifade Mountant (Thermo Fisher Scientific) and viewed with Zeiss LSM510 Upright Confocal System.

**In vivo BrdU incorporation and detection**. Mice were injected i.p. with BrdU (Life technologies) 12 h and 0.5 h before the analysis. Single-cell suspensions were prepared from harvested spleens and stained with FITC-conjugated anti-Brdu antibody (Catalog number: RUO-559619, BD Biosciences) at 4 °C following the manufacturer's instruction.

**ELISA-based assay of urinary albumin:creatinine ratio**. Mice were placed in customized spot urine cages individually for 24 h, and urine samples were collected. Urine albumin was determined using a Mouse Albumin ELISA Quantitation Kit (Catalog number: E99-134, Bethyl Laboratories) following the manufacturer's instruction. Urine creatinine was measured by Creatinine Parameter Assay Kit (R&D system) following the manufacturer's instruction.

**Administration of minicircle DNA constructs in vivo**. The minicircle DNA constructs were generated as previously described[31]. Briefly, a single isolated colony from a fresh plate was grown for the first 8 h in Luria–Bertani broth with ampicillin, inoculated into Terrific broth for further expansion and grown for 17

additional hours. Subsequently, the medium was replaced with fresh Luria–Bertani broth containing 1% L-arabinose. After adding one-half volume of fresh low salt Luria–Bertani broth (pH 8.0) containing 1% L-arabinose, the incubation temperature was increased to 37 °C and the incubation continued for two additional hours. Episomal DNA circles were prepared from bacteria using the Endofree Qiagen Megaprep plasmid purification kits (Chatsworth, CA). Hydrodynamic delivery of 8 µg MC DNA per mouse was performed via tail vein injection in a total volume of ~10% of the mouse body weight within a period of 5–7 s in all experiments. Eight-week-old mice were used and sustained high levels of targeted gene expression in vivo were achieved.

**Flow cytometry**. Cell suspensions were prepared from spleens and stained at 4 °C in PBS containing 2% FCS and 0.5% EDTA after FcγRII/III blocking. The following fluorescence antibodies were applied according to the manufacturer's instructions: anti-mouse-CD4 (Clone GK1.5), anti-mouse-CD8 (Clone 53–6.7), anti-mouse-TCRβ (Clone H57-597), anti-mouse-Ly49 (Clone 14B11), anti-mouse-CD3 (Clone 145-2C11), anti-mouse-NK1.1 (Clone PK136), anti-mouse-CD45.1 (Clone A20), anti-mouse-CD90.1 (Clone OX-7), anti-mouse/human B220 (Clone RA3-6B2), anti-human-CD56 (Clone 39D5), anti-human-CD4 (Clone A161A1), anti-human-CD8 (Clone HIT8a), anti-human-CD3 (Clone APA1/1), anti-human-TCRα/β (Clone IP26). All antibodies were from either eBioscience or BioLegend except Mouse Vβ TCR Screening Panel (BD Biosciences) and diluted according to manual from the manufacturer's website. Dead cells were excluded using Fixable Viability Dye staining (Life Technologies).

**TCR repertoire analysis**. Splenic CD4, CD8, and DN T cells were FACS sorted from a 16-week-old MRL.*lpr* mouse, and mRNAs were extracted from 1 million of each subset using Qiagen RNA isolation kit (catalog number: 74106). The Library preparation and TCR profiling were performed using Takara SMARTer Mouse TCRα/β profiling kit (catalog number: 634402). The alignment and assembly of the RNA-seq reads and the exporting of the clonotypes were conducted using MiXCR.

**Peptide:MHC II tetramer-based enrichment**. Tetramer was prepared as previous described[33,55]. Single-cell suspension was prepared. PE-conjugated tetramer was added at a concentration of 4–50 nM, depending on the saturating dose for each production batch, and the cells were incubated at room temperature for 1 h, followed by a wash in 15 ml of ice-cold sorter buffer (PBS + 2% fetal bovine serum, 0.1% sodium azide). The tetramer-stained cells were then resuspended in a volume of 0.4 ml of sorter buffer, mixed with 0.1 ml of anti-PE antibody-conjugated magnetic microbeads (Miltenyi Biotech), and incubated on ice for 20 min, followed by two washes with 10 ml of sorter buffer. The cells were then resuspended in 3 ml of sorter buffer and passed over a magnetized LS column (Miltenyi Biotech). The column was washed with sorter buffer and then removed from the magnetic field. The bound cells were obtained by pushing 5 ml of sorter buffer through the column with a plunger. The resulting enriched fractions were resuspended and was stained with a cocktail of fluorochrome-labeled antibodies[55].

**Statistics**. All results are shown as the mean ± SEM. A two-tailed, unpaired Student's *t* test was used when only two groups were compared for statistical differences. For a comparison of more than two groups, the one-way ANOVA was applied. *P* values <0.05 were considered significant.

**Study approval**. For human studies, written informed consent was obtained from all participants, and all studies were approved by the institutional review board (Committee on Clinical Investigations) at BIDMC. All animal studies were approved by the Institutional Animal Care and Use Committee at BIDMC.

**Reporting summary**. Further information on research design is available in the Nature Research Reporting Summary linked to this article.

## Data availability
TCR sequence data have been deposited in the NCBI Gene Expression Omnibus database under the primary accession code GSE149239. Source data are provided as a Source Data file. Source data are provided with this paper.

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

## Acknowledgements

This worked was supported by grants RO1 AI085567 to G.C.T. and T32 DK007199 to H.L. IEA was partly supported by a NIH/NIAMS Grant 2R01AR062173, and a National Psoriasis Foundation Translational Research grant. We would like to thank Dr. Michael Carroll, Boston Children Hospital, for his advice on Tetramer preparation.

## Author contributions

H.L. and G.C.T. designed experiments, analyzed the data, and wrote the paper. H.L., V.R.M., and A.M. performed the experiments. Z.H. performed the TCR-seq. J.M. provided U1-70:I-A^b tetramers. I.E.A. provided the minicircle vector plasmids with expert advice. S.K. procured human samples. I.E.S. provided the biopsy samples from patients with lupus nephritis. M.G.T. provided expert advice and critically reviewed the paper. All authors reviewed the paper.

## Competing interests

The authors declare no competing interests.
