## [Peer Review File · Nature Communications]

Reviewers' comments:

Reviewer #1 (Remarks to the Author):

Nature Communications manuscript NCOMMS-19-10685

Title:

Lupus milieu favors the conversion of self-reactive CD8 T cells to IL-17 producing double negative T cells

Splenic marginal zone macrophages (MZMs) are involved in establishing immune tolerance and their loss or dysfunction contributes to the progression of SLE. This important study demonstrates that loss of MZMs impairs the tolerogenic clearance of apoptotic cells and alters the serum cytokine profile which in turn provokes the generation of DN T cells from self-reactive CD8 T cells.

Increased Ki67 expression, narrowed TCR V-beta repertoire usage and diluted T-cell receptor excision circles confirmed that DN T cells from lupus-prone mice and patients with SLE patients undergo clonal proliferation and expansion in a self-antigen dependent manner. These results provide a new link between the loss of MZMs and the expansion of DN T cells and suggest new strategies to prevent the development of SLE.

Specific comments:

1. The paper should include a mechanistic diagram showing where the interactions between MZMs and CD8 T cells lies and how this leads to the expansion of IL-17-producing DN T cells in SLE.
2. Figure 3: critical data in this figure need to be supported by statistical analysis beyond statements in the figure legend. Cumulative data, such as bar charts, should be included to show the extent of overall statistically significant changes.
3. There are potentially conflicting statements in the abstract about the role of MZM in disease progression versus targeting these cells for disease prevention. These sentences should be revised or better reconciled.
4. MZMs have been implicated in clearing apoptotic debris in SLE. Although this mechanism could plausibly contribute to lupus pathogenesis (Immunol Rev. 2016 Jan;269(1):26-43), apoptosis is overall diminished in SLE patients and most animal models as evidenced by the strong impact of the Fas/lpr/CD95 mutation. In fact, this study relies on the lpr mouse as a model of SLE. While apoptosis is diminished, a pro-inflammatory form of cell death, necrosis, is increased due to defective mitochondrial ATP production in SLE (Arthritis Rheum. 2002 Jan;46(1):175-90). Since DN T cells are major source of necrotic debris in SLE (J Immunol. 2013 Sep 1;191(5):2236-46), it would be important to address whether MZM deficiency also leads to increased release of necrotic debris.
5. The relationship of MZMs to anti-inflammatory M2 macrophages should be addressed. Apparently, M2 function depends on mTORC1 (Nat Commun. 2016; 7: 13130). Since blockade of mTORC1 also prevents the CD8-DN T cell transition (Lancet. 2018 Mar 24;391(10126):1186-1196), it should be discussed if dysfunction of MZMs may be mTOR-dependent?
6. While the effect of MZMs on CD8 to DN T-cell transition is strongly supported by the studies in mice, direct evidence for involvement of this mechanism in SLE patients appears to be lacking. This relative weakness should be pointed out in the discussion, perhaps alluding the ways how this question could be addressed in the future. To this end, one wonders if surrogates of MZMs could be assessed in peripheral blood or rare splenectomy of lymph node biopsy samples of human subjects.

Reviewer #2 (Remarks to the Author):

In the manuscript the authors link the expansion of pathogenic DN T cells in lupus-prone mice with a dysfunctional marginal zone (MZ) and associated defects in apoptotic cell clearance. Major findings include 1) when MZ macrophages are depleted autoimmunity is increased, 2) there is an expansion of IL-17+ DN T cells in both the spleen and kidney, 3) loss of TGFb1 production by

myeloid cells increases T cell responses to apoptotic cell-associated antigens, and 4) skewing of TCR Vb usage is a conserved feature in human and mouse SLE (in the MRL-lpr mouse model). The paper is conceptually interesting, but there are several mechanistic links that are missing thereby reducing overall enthusiasm for the report.

Major points-

1. Several groups have reported acceleration of lupus and associated alterations in apoptotic cell-induced immunity in the absence of MZ macrophages. This includes clodronate-mediated depletion, diphtheria toxin receptor transgene expression, and spontaneous models; thus this aspect of the report is not novel. That being said, potential impact on DN T cells and the role of this cellular population in disease etiology would be impactful. Nevertheless, disruption of the MZ architecture is known to alter CD4 T cell and MZ B cell function in ways that are detrimental to peripheral tolerance. Thus it is not clear to me that the expanded DN T cells are a driver of increased autoimmunity rather than an epiphenomenon associated with amplified inflammation. It would be helpful to deplete this population or perform adoptive transfer experiments to demonstrate the pathogenicity of DN T cells in this context.

2. The authors show DN T cells expansion in response to apoptotic cell antigens. In the context of the model antigen (OVA) this might be expected, but what is much more surprising is the fact that polyclonal DN T cells expand after a similar challenge with apoptotic cells. This would suggest that either 1) nonspecific inflammation leads to expansion of DN T cells or 2) The DN T cell population contains a large proportion of self-reactive T cells. The manuscript addresses this somewhat in figure 7 which shows TCR V beta skewing to a similar degree in mice and human SLE patients. However, this level of analysis is insufficient to make any definitive assertions with regards to clonal selection or antigen specificity either in the inter-mouse or inter-species comparisons. The authors should sequence the TCR to look for similarities in CDR assemblage and allow determination of clonality. Moreover, in human SLE heterogeneity in the HLA and differences in the TCR gene segments in mice and human make a mouse /human comparison difficult. Do the authors have another way to demonstrate similarity in inter-species DN T cell selection?

3. The data suggest in the absence of MZ macrophages another phagocyte population is differentially impacted by apoptotic cell exposure driving DN T cell expansion. Given the links between DC1 dendritic cells, cross presentation, and CD8 T cell tolerance it is possible that a loss of MZ architecture leads to aberrant DC1 function driving inflammatory immunity. This could be a key pathogenesis mechanism and should be investigated.

4. Marginal zone macrophages have been reported to produce low levels of TGFb1 relative to IL-10 after apoptotic cell phagocytosis. This is in contrast to dendritic cells that preferentially produce TGFb1 after efferocytosis. Since LysM deletes in both splenic macrophages and dendritic cells the experiment as designed fails to identify the cellular source of TGFβ. The authors should address this point.

Minor points

1. Text references to figure 1c and 1d are incorrect in the results section and should be corrected.

2. There are multiple instances where FACS plots are shown but the aggregate data for the group is not shown. Bar graphs showing the mean value +/- the deviation should be included.

Question 1. *Solid experimental evidence that the novel findings in mice are relevant to the mechanism of SLE pathogenesis. This includes detailed characterization of the DN T cell antigen specificity with relation to autoimmune pathology (as per Reviewer 2 point 2). We also encourage you to provide causative evidence that this population contributes to pathogenesis by functional interrogation experiments, such as adoptive transfers of DN T cells.*

Response. Antigen-specific T cells have been implicated in the pathogenesis of SLE. Patients with SLE develop autoantibodies against DNA- and RNA-containing autoantigens. U1-70, U1-A, and U1-C, together with U1-RNA and the seven Smith proteins compose the U1-small nuclear ribonucleoprotein (U1-snRNP) complex. It has been reported that the frequency of Th17 cells specific for U1-70-snRNP correlates with anti-U1-70 autoantibody production and disease severity. To characterize the antigen specificity of DN T cells in lupus-prone mice, we have generated IL-17 GFP-MRL/lpr mice and fluorescence was used to track IL-17⁺ T cells *in vivo*. Splenic T cells from IL-17 GFP-MRL/lpr mice were co-cultured with irradiated antigen presenting cells pre-loaded with U1-70 peptide in the presence of brefeldin A. Production of IL-17 and IFN γ were examined and IL-17 but not IFN γ production was detected among GFP⁺ CD4 T cells and GFP⁺ DN T cells, which indicated that a portion of IL-17⁺ DN T cells can recognize and respond to self-antigen U1-70 (**Fig. 4g**). Furthermore, U1-70:I-A^b tetramers were applied to identify auto-reactive T cells from IL-17 GFP-MRL/lpr mice which recognize U1-70 in MHC class II dependent manner. Consistent with our data that DN T cells were derived from CD8 but not CD4 T cells, almost no DN T cells could bind to the U1-70:I-A^b tetramers even a great portion of GFP⁺ CD4 T cells were positive for U1-70:I-A^b tetramers (**Fig. 4h**). Taken together, our data suggested that U1-70-specific DN T cells respond to U1-70 and produce IL-17A in MHC class I but not class II dependent manner.

We next examined the pathogenic role of DN T cells in lupus, specifically their ability to provide help to B cells to produce autoantibody *in vivo*. Splenic CD4, CD8 and DN T cells were enriched from B6/lpr mice and transferred into *Rag1*^{-/-} recipients which received also purified B cells also from B6/lpr mouse spleens one day earlier. Autoantibodies in the circulation were examined one month after transfer. The group which received DN T cells and B cells displayed higher titers of circulating autoantibodies and more renal immune complex deposition compared to groups which received B cells alone or B cells and CD8 T cells (**Supplemental Fig. 13**), which is consistent with our previous reports that DN T cells from SLE patients can augment the production of pathogenic anti-DNA autoantibodies *in ex vivo* experiments using peripheral blood cells from patients with SLE (S Shivakumar, GC Tsokos, S. Datta, *J Immunol.* 1989).

Question 2. *Methodology. It remains to be demonstrated that the approaches to manipulate phagocytes (Cre lines and clodronate depletion) are specific and selective to MZM. To address this, the impact on other myeloid and DC subsets has to be measured for each experimental model. If other populations are affected (Reviewer 2 points 3,4), the contribution of the other cell subsets has to be ruled out and the role of MZM has to be confirmed by other lines of evidence.*

Response. The dose of clodronate liposome for targeting delivery to MZMs has been optimized using fluoroliposome in our previous report (Li H et al, *J Immunol.* 2013). To confirm that the appropriate dose of clodronate liposome (CL) preferentially targeted marginal zone macrophages with minimal impact on other splenic macrophages or dendritic cells, FACS analysis was applied to enumerate the percentage of different myeloid cells in the spleens of mice treated with indicated dose of CL. In supplemental Figure 2, the flow plots and bar graphs clearly demonstrate the reduction of marginal zone macrophages but splenic CD11b^{high} neutrophils and CD11b^{lo}F4/80⁺ remained intact after CL administration. Confocal images are provided to determine the integrity of MZM in the spleen and, as expected, loss of marginal zone macrophage ring was observed, however the CD68⁺ red pulp macrophages were well preserved (**Fig. 5e**). Taken together, our results confirmed the selective deletion of marginal zone macrophages by the indicated dose of clodronate liposome.

We agreed with the reviewer that lysozyme expression was not restricted to marginal zone macrophages, it was detected in other macrophages and dendritic cells. The phenotype observed in *Lyz2-cre x Tgfβ1^{fl/fl}* mice could be explained as the defective TGFβ1 production in other myeloid cells, for example CD8⁺ dendritic cells, which have been proposed delivering apoptotic antigens to T cells for tolerance induction. To address this concern, we injected CFSE labelled apoptotic cells into mice with or without MZM deletion by CL and the clearance of administrated apoptotic cells were tracked by confocal imaging. The numbers of CFSE⁺ cells, which were located predominately in the MZ, peaked at 5–15 minutes and the majority of them was phagocytosed at 20 min. Of note, the leakage of apoptotic cell debris into the follicular area was observed in the absence of MZMs, which makes the debris accessible to other myeloid cells like CD8⁺ dendritic cells, located inside the follicles (**Fig. 5f**).

The spleen is the main filter of circulating apoptotic cells. We next interrogated whether MZMs were the main cellular sources of TGFβ during the clearance of endogenous circulating apoptotic cells. Different population myeloid cells were FACS sorted (including marginal zone macrophages, red pulp macrophages, CD4⁺ dendritic cells and CD8⁺ dendritic cells) from mice under different treatments. Consistent with our confocal imaging results above, with intact marginal zone barrier, marginal zone macrophages expressed much higher amounts of TGFβ1 compared to other macrophages and dendritic cells but absence of MZMs led to the activation of other myeloid cells by administrated exogenous apoptotic cells (**Fig. 5c**). In contrast to red pulp macrophages and CD4⁺ dendritic cells, which produce various inflammatory cytokines, CD8⁺ dendritic cells produce high levels of the immune suppressive cytokine, TGFβ1, in the absence of MZM barrier (**Fig. 5d**). Taken together, our results strongly suggest that under naïve status, marginal zone macrophages are essential for the initiation of immune tolerance to apoptotic cells by producing TGFβ1.

Specific Criticisms by Reviewers.

Reviewer A.

Question 1. *The paper should include a mechanistic diagram showing where the interactions between MZMs and CD8 T cells lies and how this leads to the expansion of IL-17-producing DN T cells in SLE.*

Response. A mechanistic diagram has been added as requested (**Supplemental Fig. 21**).

Question 2. *Figure 3: critical data in this figure need to be supported by statistical analysis beyond statements in the figure legend. Cumulative data, such as bar charts, should be included to show the extent of overall statistically significant changes.*

Response. Quantifications have been provided in **Supplemental Fig. 6**.

Question 3. *There are potentially conflicting statements in the abstract about the role of MZM in disease progression versus targeting these cells for disease prevention. These sentences should be revised or better reconciled.*

Response. We thank the reviewer for this important comment. The indicated sentence has been revised.

Question 4. *MZMs have been implicated in clearing apoptotic debris in SLE. Although this mechanism could plausibly contribute to lupus pathogenesis (Immunol Rev. 2016 Jan;269(1):26-43), apoptosis is overall diminished in SLE patients and most animal models as evidenced by the strong impact of the Fas/lpr/CD95 mutation. In fact, this study relies on the lpr mouse as a model of SLE. While apoptosis is diminished, a*

pro-inflammatory form of cell death, necrosis, is increased due to defective mitochondrial ATP production in SLE (Arthritis Rheum. 2002 Jan;46(1):175-90). Since DN T cells are major source of necrotic debris in SLE (J Immunol. 2013 Sep 1;191(5):2236-46), it would be important to address whether MZM deficiency also leads to increased release of necrotic debris.

Response. We thank the reviewer for the comment. In *lpr* mice, the defective lymphocyte apoptosis mediated by Fas/CD95 has been assumed to be the underlying cause of the disease. The abnormalities in apoptosis might lead to increased rates of other forms of programmed cell death like necrosis. No matter how cells die, either through apoptosis or necrosis, they must be properly removed to prevent autoimmunity. Obviously, MZM deficiency leads to defective elimination of autoantigens and is responsible for the pathological findings in lupus. The indicated references have been included in the discussion.

Question 5. *The relationship of MZMZs to anti-inflammatory M2 macrophages should be addressed. Apparently, M2 function depends on mTORC1 (Nat Commun. 2016; 7: 13130). Since blockade of mTORC1 also prevents the CD8-DN T cell transition (Lancet. 2018 Mar 24;391(10126):1186-1196), it should be discussed if dysfunction of MZMs may be mTOR-dependent?*

Response. Sirolimus treatment leads to a progressive improvement in SLE disease activity which is closely associated with expansion of CD4⁺CD25⁺FoxP3⁺ regulatory T cells and reduction of interleukin-17 producing CD4⁺CD8⁻ double-negative T cells (Lancet, Lai ZW et al, 2018). Lamtor1-mediated activation of mTORC1 is required for M2 polarization, however the allosteric mTORC1 inhibitor rapamycin did not block the polarization of M2 macrophages (Nat Commun. Kimura T et al, 2016). The therapeutic effects *in vivo* might be attributed to direct modulation of the mammalian target of rapamycin (mTOR) in T cells. It has been shown that LXR transcriptional activity depends on mTOR in M2 macrophages and interestingly, LXR is the key of the development fate decision for marginal zone macrophages, which suggested that dysfunction of MZMs in SLE might be mTOR-dependent. Related discussion has been expanded.

Question 6. *While the effect of MZMs on CD8 to DN T-cell transition is strongly supported by the studies in mice, direct evidence for involvement of this mechanism in SLE patients appears to be lacking. This relative weakness should be pointed out in the discussion, perhaps alluding the ways how this question could be addressed in the future. To this end, one wonders if surrogates of MZMs could be assessed in peripheral blood or rare splenectomy of lymph node biopsy samples of human subjects.*

Response. Immunohistological studies of the spleens from lupus and unaffected subjects have revealed a significant reduction of the marginal zone macrophage layer. However, additional studies of human SLE are warranted to prove that in SLE, abnormalities in marginal zone barrier occur prior to the expansion of DN T cells. Since the circulating monocytes function as MZM precursors, it will be interesting to explore whether the central defects underlying the gradual changes in MZMs start from the circulating precursors. The discussion has been expanded as suggested.

Reviewer #2 (Remarks to the Author):

Major points-

Question 1. *Several groups have reported acceleration of lupus and associated alterations in apoptotic cell-induced immunity in the absence of MZ macrophages. This includes clodronate-mediated depletion, diphtheria toxin receptor transgene expression, and spontaneous models; thus this aspect of the report is not novel. That being said, potential impact on DN T cells and the role of this cellular population in disease etiology would be impactful. Nevertheless, disruption of the MZ architecture is known to alter CD4 T cell and MZ B cell function in ways that are detrimental to peripheral tolerance. Thus it is not clear to me that the expanded DN T*

cells are a driver of increased autoimmunity rather than an epiphenomenon associated with amplified inflammation. It would be helpful to deplete this population or perform adoptive transfer experiments to demonstrate the pathogenicity of DN T cells in this context.

Response. Following the suggestion from the reviewer, splenic DN T cells were purified from B6/*lpr* mice and transferred into *Rag1*^{-/-} recipients which received purified B cells one day earlier. Autoantibodies in the circulation were examined one month after transfer. The recipients displayed much higher titers of circulating autoantibodies and more renal immune complex deposition compared to those receiving either B cells alone (**Supplemental Fig. 13**), which is consistent with our previous report that DN T cells from SLE patients can augment the production of pathogenic autoantibodies by B cells *in vitro* (S Shivakumar, GC Tsokos, S Datta, *J Immunol.* 1989).

Question 2. The authors show DN T cells expansion in response to apoptotic cell antigens. In the context of the model antigen (OVA) this might be expected, but what is much more surprising is the fact that polyclonal DN T cells expand after a similar challenge with apoptotic cells. This would suggest that either 1) nonspecific inflammation leads to expansion of DN T cells or 2) The DN T cell population contains a large proportion of self-reactive T cells. The manuscripts addresses this somewhat in figure 7 which shows TCR V beta skewing to a similar degree in mice and human SLE patients. However, this level of analysis is insufficient to make any definitive assertions with regards to clonal selection or antigen specificity either in the inter-mouse or inter-species comparisons. The authors should sequence the TCR to look for similarities in CDR assemblage and allow determination of clonality. Moreover, in human SLE heterogeneity in the HLA and differences in the TCR gene segments in mice and human make a mouse /human comparison difficult. Do the authors have another way to demonstrate similarity in inter-species DN T cell selection?

Response. Following the suggestion from the reviewer, we sequenced the TCR beta chains of splenic CD4, CD8 and DN T cells from MRL.*lpr* mouse. Of note, in the MRL.*lpr* mouse, the higher proportion of top 10 clonotypes in DN T cells suggested selective expansion of some clonotypes of DN T cells (**Supplemental Table 1, Supplemental Fig. 19**). Consistent with the flow data, increased clonal space occupation by Vβ5 clonotypes was observed in DN T cells from MRL.*lpr* mice (**Supplemental Fig. 20, Supplemental Table 2**). Antigen-specific T cells have been implicated in the pathogenesis of SLE. Patients with SLE develop autoantibodies against DNA- and RNA- containing autoantigens including U1-70, U1-A, and U1-C, which together with U1-RNA and the seven Smith proteins compose the U1-small nuclear ribonucleoprotein (U1-snRNP) complex. The frequency of Th17 cells specific for U1-70-snRNP correlates with anti-U1-70 autoantibody production and disease severity. To characterize the antigen specificity of DN T cells in lupus-prone mice, we generated IL-17 GFP-B6/*lpr* mice and fluorescence was used to track IL-17⁺ T cells *in vivo*. Next, we cultured the splenic T cells from IL-17 GFP-B6/*lpr* mice with irradiated antigen presenting cells loaded with the U1-70 peptide in the presence of brefeldin A. Production of IL-17 and IFNγ was examined and as expected, IL-17 but not IFNγ production was detected in GFP⁺ CD4 T cells and GFP⁺ DN T cells (**Fig. 4g**), which indicated a portion of IL-17⁺ DN T cells could respond to self-antigen U1-70 by producing IL-17. Furthermore, U1-70:I-A^b tetramers were applied to identify auto-reactive T cells from IL-17 GFP-MRL/*lpr* mice which recognize U1-70 in MHC class II dependent manner (**Fig. 4h**). Consistent with our data that DN T cells derive from CD8 but not CD4 T cells, almost no DN T cells could bind U1-70:I-A^b tetramers even a great proportion of GFP⁺ CD4 T cells are positive for U1-70:I-A^b tetramers. Taken together, our data suggest that U1-70-specific DN T cells respond to U1-70 and produce IL-17A in MHC class I but not class II-dependent manner.

Question 3. The data suggest in the absence of MZ macrophages another phagocyte population is differentially

impacted by apoptotic cell exposure driving DN T cell expansion. Given the links between DC1 dendritic cells, cross presentation, and CD8 T cell tolerance it is possible that a loss of MZ architecture leads to aberrant DC1 function driving inflammatory immunity. This could be a key pathogenesis mechanism and should be investigated.

Response. Marginal zone macrophages (MZMs) act as a barrier to entry of circulating apoptotic debris into the follicles of secondary lymphoid organs. The dose of clodronate liposome (CL) that we applied *in vivo* preferentially deleted MZMs with minimal effects on other splenic macrophages or dendritic cells, a fact that was confirmed by the cytometry analysis in **Supplemental Fig. 2**. To test which population of myeloid cells processes apoptotic cells in the absence of MZMs, CFSE labeled apoptotic cells were administered into mice after MZM depletion and the capture of apoptotic cell debris was tracked by CFSE intensity. Macrophages located in the red pulp are known as red pulp macrophages (RPMs) and they can be rapidly mobilized to leave the spleen and assist in fighting against infections because they express higher levels of MHC II and costimulatory molecules like CD80, CD86 (*Immunol Rev.* 2016 Jan;269(1):26-43). Interestingly, absence of MZMs results in retention of apoptotic cell (AC) debris within the marginal zone (MZ) and increased loading of apoptotic cell debris on the CD68⁺ red pulp macrophages as indicated by the intensity of CFSE compared to other myeloid cells especially dendritic cells (**Fig. 5g**). The confocal imaging was also applied to examine the migration behavior of different myeloid cells in the absence of MZMs after CFSE labeled apoptotic cells were injected. Of note, in the absence of MZMs, administration of apoptotic cells leads to the translocation of CD68⁺ red pulp macrophages to the follicle (**Fig. 5e**), which indicates that these macrophages may be able of translocating the antigens to the follicular zone to activate T and B cells.

Question 4. *Marginal zone macrophages have been reported to produce low levels of TGFb1 relative to IL-10 after apoptotic cell phagocytosis. This is in contrast to dendritic cells that preferentially produce TGFb1 after efferocytosis. Since LysM deletes in both splenic macrophages and dendritic cells the experiment as designed fails to identify the cellular source of TGFβ. The authors should address this point.*

Response. We agree with the reviewer that Lysozyme expression is not restricted to marginal zone macrophages, and it is detected in other macrophages and dendritic cells. The phenotype observed in *Lyz2-cre x Tgfβ1^{fl/fl}* mice can be explained as the defective TGFβ1 production in other myeloid cells, for example CD8⁺ dendritic cells, which have been proposed to be able to deliver apoptotic antigens to T cells for tolerance induction. The spleen is the main filter of circulating apoptotic cells and we first asked whether MZMs were the main cellular sources of TGFβ during the clearance of circulating apoptotic cells. Apoptotic thymocytes were administered into mice pre-treated with or without CL for MZM depletion. Different populations of myeloid cells including marginal zone macrophages, red pulp macrophages, CD4⁺ dendritic cells and CD8⁺ dendritic cells were sorted out by FACS 30 minutes after administration and the production of various cytokines including *Tgfβ1* was examined by qRT-PCR. As expected, with intact marginal zone barrier, marginal zone macrophages expressed much higher levels of TGFβ1 compared to other myeloid cells after exposure to apoptotic cells. However, in the absence of MZMs, activation of other myeloid cells by exogenous apoptotic cells was observed as indicated by the elevation of the proinflammatory cytokines including IL-6 and IL-23 especially in red pulp macrophages (**Fig. 5d**). Of note, in contrast to red pulp macrophages and CD4⁺ dendritic cells, which produce pro-inflammatory cytokines, CD8⁺ dendritic cells produce high level of immune suppressive cytokine TGFβ1 only when the MZM barrier was disrupted which is consistent with previous reports that CD8⁺ dendritic cells have the ability to induce tolerance by producing TGFβ1 (**Fig. 5d**). Collectively, our results strongly suggest that under naïve conditions, marginal zone macrophages are the main cellular source of TGFβ1 maintaining immune tolerance to apoptotic cells.

Minor points

Question 1. *Text references to figure 1c and 1d are incorrect in the results section and should be corrected.*

Response. We thank the reviewer for this comment. The text has been corrected.

Question 2. *There are multiple instances where FACS plots are shown but the aggregate data for the group is not shown. Bar graphs showing the mean value +/- the deviation should be included.*

Response. The bar graphs have been added as suggested (**Fig. 1d, Supplemental Fig. 2b, 3, 6, 8**).

We greatly appreciate the important suggestions which were provided by you and reviewers and look forward to your positive response.

Sincerely,

George C. Tsokos, MD

REVIEWERS' COMMENTS:

Reviewer #1 (Remarks to the Author):

The paper has been extensively revised in response to the initial reviews. The results demonstrate a novel mechanism of lupus development implicating marginal zone macrophages as protectors in disease pathogenesis through blocking the expansion of pro-inflammatory DN T cells with remarkable potency. MZM cells may serve as a new tool for treatment in SLE.

Reviewer #2 (Remarks to the Author):

The authors have sufficiently answered my critique. I don't have further concerns.